# Design and Synthesis of Flavonoidal Ethers and Their Anti-Cancer Activity In Vitro

**DOI:** 10.3390/molecules24091749

**Published:** 2019-05-06

**Authors:** Lu Jin, Meng-Ling Wang, Yao Lv, Xue-Yi Zeng, Chao Chen, Hai Ren, Heng Luo, Wei-Dong Pan

**Affiliations:** 1State Key Laboratory of Functions and Applications of Medicinal Plants, Guizhou Medical University, Guiyang 550014, China; m18352807507@163.com (L.J.); WANGMENGLING417@163.com (M.-L.W.); Xueyizeng@126.com (X.-Y.Z.); cc283818640@163.com (C.C.); renh0206@163.com (H.R.); 2The Key Laboratory of Chemistry for Natural Products of Guizhou Province and Chinese Academy of Sciences, Guiyang 550014, China; 3Bijie Medical College, Bijie 551700, China; Lvyao0918@163.com

**Keywords:** flavonoids, allene, E-stereoselective, regioselective, anti-cancer activity

## Abstract

Flavonoids are well-characterized polyphenolic compounds with pharmacological and therapeutic activities. However, most flavonoids have not been developed into clinical drugs, due to poor bioavailability. Herein, we report a strategy to increase the drugability of flavonoids by constructing C(sp^2^)-O bonds and stereo- as well as regioselective alkenylation of hydroxyl groups of flavonoids with ethyl-2,3-butadienoate allenes. Twenty-three modified flavonoid derivatives were designed, synthesized, and evaluated for their anti-cancer activities. The results showed that compounds **4b**, **4c**, **4e**, **5e**, and **6b** exhibited better in vitro inhibitory activity against several cancer cell lines than their precursors. Preliminary structure–activity relationship studies indicated that, in most of the cancer cell lines evaluated, the substitution on position 7 was essential for increasing cytotoxicity. The results of this study might facilitate the preparation or late-stage modification of complex flavonoids as anti-cancer drug candidates.

## 1. Introduction

Consisting of more than 9000 compounds, flavonoids represent the most widely distributed polyphenols in nature [1,2]. In addition, flavonoids are ubiquitous in some medicinal plants and herbal remedies used in traditional medicine around the world, especially in China [2,3,4,5,6]. The characteristic skeleton of flavonoids is a phenylbenzopyrone moiety (C6-C3-C6), which could be further categorized according to the saturation level and the presence or absence of the central pyran ring. Many studies have confirmed that flavonoids are naturally occurring, pharmacologically active molecules. However, most flavonoids have not been developed as clinical drugs because of poor bioavailability (less than 5%), toxicity, and induction or inhibition of some metabolic enzymes [7]. To overcome these problems, structural modifications of the flavonoid skeletons have attracted great interest [8].

Allenes are highly reactive and have been used extensively in organic chemistry [9,10,11,12,13,14,15], due to the electronic or steric effects and their 1,2-diene functionality. The transition-metal catalyzed chemistry of allenes has become a recent focus of research [16,17]. We proposed that carbopalladation of allenes would provide a highly convenient method for the formation of a conjugated aryl ether, following *β*-H elimination by the introduction of a π-allyl palladium species.

In this research, we have developed a friendly and straightforward protocol to access various novel flavonoidal ethers by employing allenes as important multifunctional modules. Notably, both electrophilic groups and aliphatic chains can be embedded in the flavonoid scaffold smoothly with the above mentioned protocol. Therefore, more than 20 flavonoidal ethers were obtained successfully and evaluated for anti-cancer activities.

## 2. Materials and Methods

### 2.1. Instruments and Materials

^1^H- and ^13^C-NMR spectra were obtained on a 400 MHz (Varian, Inc) or 500 MHz (WIPM, China) spectrometer in CDCl_3_ or DMSO-*d_6_* (tetramethylsilane (TMS) as internal standard). Chemical shifts (*δ*) were expressed in parts per million (ppm), relative to TMS (0 ppm). High resolution mass spectra (HRMS) were recorded on a ThermoFisher QE Focus apparatus. All the solvents were dried using standard methods and distilled before being used. Reagents and solvents were purchased from commercial sources. Solvents were purified according to the guidelines in the Purification of Laboratory Chemicals.

### 2.2. General Procedure for the Synthesis of Flavonoidal Ether Derivatives

#### 2.2.1. Synthesis of Allenes

Allenes were synthesized according to previously described protocols [16,17,18].

#### 2.2.2. Optimization of Reaction Conditions

We selected chrysin **2a** and ethyl ester allene **1a** as the model substrates in this study and carried out extensive screening of the reaction conditions. In an initial attempt, a variety of factors including types of Pd catalysts, bases, solvents, and ligands were examined. After extensive optimization, the product (**3a**) was obtained with an isolated yield of 41.7% in the presence of 10 mol% of PPh_3_/Pd(dba)_2_ as a catalyst, K_2_CO_3_ (2.0 equiv.) as a base, in a solution of MeCN under argon at 80 °C (Table 1, entry 1). Further investigations of the solvent indicated that the yield could be slightly improved to 43.0% by replacing MeCN with DMF (Table 1, entry 6). Furthermore, absence of Pd catalysts and ligands resulted in lower yield (Table 1, entry 14). None of the desired product was obtained when the base was abstracted away (Table 1, entries 15 and 16). Interestingly, only 11.2% yield of the expected product was obtained at lower temperatures (Table 1, entry 17). It was worth noting that **3a** was the only isomer isolated, indicating a highly regio- and stereoselective process, in which the steric effect of the carbonyl or the hydrogen bond effect between the carbonyl and oxygen atom at position 7 may be critical for the highly regio- and stereoselective process.

#### 2.2.3. General Procedure for the Synthesis of Target Derivatives **3a**–**3i**, **4a**–**4g**, **5a**–**5e**

Chrysin **2a** (127.12 mg, 0.5 mmol), Pd(dba)_2_ (5.75 mg, 0.01 mmol), K_2_CO_3_ (138.21 mg, 1.0 mmol), PPh_3_ (13.11 mg, 0.05 mmol), allene (112.13 mg, 1.0 mmol), and anhydrous DMF (5 mL) were added to a 30 mL tube under argon, which was then sealed. The resulting mixture was stirred at 80 °C for 12 h. After completion, the reaction solution was cooled down to room temperature before 80 mL water was added, followed by the addition of 1 N HCl aqueous solution at 0 °C, until the pH value reached 7. Ethyl acetate (100 mL) was added to the reaction solution and the organic layer was washed sequentially with a large amount of water (4 × 200 mL), and then brine, before being dried over anhydrous Na_2_SO_4_. The solvent was evaporated under reduced pressure. The crude product was purified by silica gel column chromatography (petroleum ether–ethylacetate, 5:1) to give **3a** (78.8 mg, 43% yield). Compounds **3b**–**3i, 4a**–**4g,** and **5a**–**5e** were prepared according to similar procedures in which the double-substitution is a byproduct. The products **4a**–**4g** were generated in a shorter time compared with the other products (within 10 h).

#### 2.2.4. General Synthetic Procedure for **6a**–**6b** Target Derivatives

The reaction with 1.0 mmol scale was carried out to assess the scalability of flavonoidal ether formation (Scheme 1, **4a**). Under standard conditions, the yield was lower than that of the small-scale reaction. The reaction solution contained the following components: **4a** (62.8 mg, 0.02 mmol), 1 mL methanol, and 1 N sodium hydroxide solution (2.0 equiv). The solution was then stirred at 60 °C for 5 min and 1 mL water was added before heating at 60 °C for 12 h. After completion, the reaction solution was cooled down to room temperature and 40 mL water as well as 30 mL ethyl acetate was added sequentially. Aqueous HCl (1 N) was added to the water layer at 0 °C until the pH value reached 4. Other post-processing procedures were similar to those performed to synthesize **4a**. The desired product **6a** was finally obtained (41.2 mg, 61% yield). The product **6b** was prepared according to the same general procedure.

*Ethyl (E)-3-((5-hydroxy-4-oxo-2-phenyl-4H-chromen-7-yl) oxy) but-2-enoate* (**3a**); yellow powder; yield 43.0%; HRMS(ESI): calcd. C_21_H_17_O_6_, *m*/*z* 365.1031 [M − H]^−^, found 365.1025; ^1^H-NMR (CDCl_3_, 500 MHz) *δ*(ppm): 7.88 (d, *J* = 7.0 Hz, 2H), 7.59–7.51 (m, 3H), 6.73 (s, 1H), 6.68 (d, *J* = 2.0 Hz, 1H), 6.49 (d, *J* = 2.0 Hz, 1H), 5.16 (s, 1H), 4.12 (q, *J* = 7.0 Hz, 2H), 2.49 (s, 3H), 1.23 (t, *J* = 7.0 Hz, 3H); ^13^C-NMR (CDCl_3_, 125 MHz) *δ*(ppm): 182.9, 170.5, 167.1, 164.7, 162.6, 159.4, 157.4, 132.3, 131.0, 129.3, 126.5, 108.6, 106.3, 104.8, 100.0, 99.8, 60.0, 18.2, 14.4. 

*Ethyl (E)-3-((5,6-dihydroxy-4-oxo-2-phenyl-4H-chromen-7-yl) oxy) but-2-enoate* (**3b**); yellow powder; yield 43.9 %; HRMS(ESI): calcd. C_21_H_17_O_7_, *m*/*z* 381.0980 [M − H]^−^, found 381.0970; ^1^H-NMR (DMSO-*d*_6_, 400 MHz) *δ*(ppm): 13.02 (s, 1H), 8.04 (d, *J* = 7.2 Hz, 2H), 7.59–7.52 (m, 3H), 6.97 (s, 1H), 6.69 (s, 1H), 4.80 (s, 1H), 3.97 (q, *J* = 7.2 Hz, 2H), 2.40 (s, 3H), 1.09 (t, *J* = 7.2 Hz, 3H); ^13^C-NMR (DMSO-*d*_6_, 100 MHz) *δ*(ppm): 182.3, 170.7, 166.6, 163.7, 156.7, 154.2, 152.4, 132.2, 130.7, 129.2, 126.5, 123.8, 105.0, 104.4, 94.8, 93.3, 59.2, 17.6, 14.2. 

*Diethyl 3,3′-((5-hydroxy-4-oxo-2-phenyl-4H-chromene-6,7-diyl) bis (oxy)) (2E,2′E)-bis (but-2-enoate)* (**3c**); yellow powder; yield 29.3%; HRMS(ESI): calcd. C_27_H_27_O_9_, *m*/*z* 495.1650 [M + H]^+^, found 495.1647; ^1^H-NMR (DMSO-*d*_6_, 500 MHz) *δ*(ppm): 11.76 (s, 1H), 8.04 (d, *J* = 6.0 Hz, 2H), 7.60–7.57 (m, 3H), 7.16 (s, 1H), 6.78 (s, 1 H), 4.77 (s, 1H), 4.63 (s, 1H), 4.01–3.94 (m, 4H), 2.44 (s, 3H), 2.39 (s, 3H), 1.16–1.03 (m, 6H); ^13^C-NMR (DMSO-*d*_6_, 125 MHz) *δ*(ppm): 174.7, 171.6, 170.4, 166.3, 166.2, 161.0, 155.4, 155.3, 143.0, 131.7, 131.3, 130.8, 129.1, 126.3, 109.8, 107.5, 102.7, 94.5, 93.9, 59.3, 59.1, 17.6, 17.3, 14.1. 

*Ethyl (E)-3-(5-(5,7-dihydroxy-4-oxo-4H-chromen-2-yl)-2-methoxyphenoxy)-2-methylacrylate* (**3d**); yellow powder; yield 25.9%; HRMS(ESI): calcd. C_22_H_19_O_8_, *m*/*z* 411.1085 [M − H]^−^, found 411.1078; ^1^H-NMR (DMSO-*d_6_*, 500 MHz) *δ*(ppm): 12.87 (s, 1H), 10.88 (s, 1H), 8.03 (dd, *J* = 9.0, 2.5 Hz, 1H), 7.87 (d, *J* = 2.5 Hz, 1H), 7.33 (d, *J* = 9 Hz, 1H), 6.95 (s, 1H), 6.52 (d, *J* = 2.0 Hz, 1H), 6.20 (d, *J* = 2.0 Hz, 1H), 4.66 (s, 1H), 3.99 (q, *J* = 7.0 Hz, 2H), 3.88 (s, 3H), 2.43 (s, 3H), 1.10 (t, *J* = 7.0 Hz, 3H); ^13^C-NMR (DMSO-*d_6_*, 125 MHz) *δ*(ppm): 181.8, 171.6, 166.3, 164.3, 162.1, 161.4, 157.4, 153.9, 141.1, 126.0, 123.6, 121.1, 113.8, 104.3, 103.8, 99.0, 94.6, 94.2, 59.2, 56.3, 17.7, 14.1. 

*Ethyl (E)-3-((2-(3-(((E)-4-ethoxy-4-oxobut-2-en-2-yl) oxy)-4-methoxyphenyl)-5-hydroxy-4-oxo-4H-chromen-7-yl) oxy) but-2-enoate* (**3e**); yellow powder; yield 21.2%; HRMS(ESI): calcd. C_28_H_29_O_10_, *m*/*z* 525.1755 [M + H]^+^, found 525.1759; ^1^H-NMR (DMSO-*d*_6_, 500 MHz) *δ*(ppm): 12.94 (s, 1H), 8.05 (dd, *J* = 8.5, 2.0 Hz, 1H), 7.86 (d, *J* = 2.0 Hz, 1H), 7.32 (d, *J* = 8.5 Hz, 1H), 7.05 (s, 1H), 6.95 (d, *J* = 2.0 Hz, 1H), 6.51 (d, *J* = 2.0 Hz, 1H), 5.05 (s, 1H), 4.66 (s, 1H), 4.05–3.95 (m, 4H), 3.90 (s, 3H), 2.42 (d, *J* = 3.5 Hz, 6H), 1.21–1.10 (m, 6H). ^13^C-NMR (DMSO-*d*_6_, 125 MHz) *δ*(ppm): 181.8, 171.6, 166.3, 164.3, 162.1, 161.4, 157.4, 153.9, 141.1, 126.0, 123.6, 121.1, 113.8, 104.3, 103.8, 99.0, 94.6, 94.2, 59.2, 56.3, 39.7, 39.5, 39.4, 17.7, 14.1.

*Ethyl (E)-3-((5-hydroxy-2-(4-hydroxyphenyl)-4-oxo-4H-chromen-7-yl) oxy) but-2-enoate* (**3f**); yellow powder; yield 30.3%; HRMS(ESI): calcd. C_21_H_17_O_7_, *m*/*z* 381.0980 [M − H]^−^, found 381.0986; ^1^H-NMR (DMSO-*d*_6_, 400 MHz) *δ*(ppm): 13.10 (s, 1H), 10.45 (s, 1H), 7.97 (d, *J* = 8.0 Hz, 2H), 6.98–6.91 (m, 4H), 6.56 (s, 1H), 5.05 (s, 1H), 4.03 (q, *J* = 7.2 Hz, 2H), 2.42 (s, 3H), 1.14 (t, *J* = 7.2 Hz, 3H); ^13^C-NMR (DMSO-*d*_6_, 100 MHz) *δ*(ppm): 182.3, 170.5, 166.1, 164.7, 161.6, 161.5, 158.4, 156.8, 128.8, 120.8, 116.0, 107.9, 104.0, 103.4, 100.2, 98.7, 59.4, 17.7, 14.1. 

*Ethyl (E)-3-(4-(5,7-dihydroxy-4-oxo-4H-chromen-2-yl) phenoxy) but-2-enoate* (**3g**); yellow powder; yield 28.3%; HRMS(ESI): calcd. C_21_H_17_O_7_, *m*/*z* 381.0980 [M − H]^−^, found 381.0976; ^1^H-NMR (DMSO-*d_6_*, 400 MHz) *δ*(ppm): 12.82 (s, 1H), 10.96 (s, 1H), 8.17 (d, *J* = 8.8 Hz, 2H), 7.34 (d, *J* = 8.4 Hz, 2H), 6.99 (s, 1H), 6.52 (d, *J* = 2.0 Hz, 1H), 6.22 (d, *J* = 2.0 Hz, 1H), 4.79 (s, 1H), 4.01 (q, *J* = 7.2 Hz, 2H), 2.45 (s, 3H), 1.12 (t, *J* = 7.2 Hz, 3H); ^13^C-NMR (DMSO-*d_6_*, 100 MHz) *δ*(ppm): 181.9, 171.7, 166.2, 164.5, 162.4, 161.5, 157.5, 155.6, 128.8, 128.3, 122.2, 105.3, 103.9, 99.1, 96.9, 94.2, 59.3, 17.8, 14.2.

*Ethyl 2-(5-(5,7-dihydroxy-4-oxo-4H-chromen-2-yl)-2-methylbenzo [d] [1,3] dioxol-2-yl) acetate* (**3h**); yellow powder; yield 30.1%; HRMS(ESI): calcd. C_21_H_17_O_8_, *m*/*z* 397.0929 [M − H]^−^, found 397.0923; ^1^H-NMR (CDCl_3_, 400 MHz) *δ*(ppm): 12.62 (s, 1H), 7.43 (dd, *J* = 8.4, 1.6 Hz, 1H), 7.26 (d, *J* = 1.2 Hz, 1H), 6.88 (d, *J* = 8.4 Hz, 1H), 6.52 (s, 1H), 6.49 (d, *J* = 1.2 Hz, 1H), 6.38 (d, *J* = 1.6 Hz, 1H), 4.20 (q, *J* = 7.2 Hz, 2H), 3.06 (s, 2H), δ 1.88 (s, 3H), 1.25 (t, *J* = 7.2 Hz, 3H); ^13^C-NMR (CDCl_3_, 100 MHz) *δ*(ppm): 182.5, 168.3, 164.1, 164.0, 162.1, 157.9, 150.5, 148.1, 124.8, 121.6, 117.6, 108.9, 106.4, 104.9, 104.0, 100.0, 94.6, 61.4, 44.0, 25.0, 14.1. 

*Ethyl 2-(2-methyl-5-(3,5,7-trihydroxy-4-oxo-4H-chromen-2-yl) benzo [d] [1,3] dioxol-2-yl) acetate* (**3i**); yellow powder; yield 30.1%; HRMS(ESI): calcd. C_21_H_19_O_9_, *m*/*z* 415.1024 [M + H]^+^, found 415.1018; ^1^H-NMR (DMSO-*d_6_*, 500 MHz) *δ*(ppm): 12.40 (s, 1H), 10.83 (s, 1H), 9.57 (s, 1H), 7.74 (dd, *J* = 8.0, 1.5 Hz, 1H), 7.62 (d, *J* = 1.5 Hz, 1H), 7.03 (d, *J* = 8.0 Hz, 1H), 6.47 (d, *J* = 2.0 Hz, 1H), 6.19 (d, *J* = 2.0 Hz, 1H), 4.02 (q, *J* = 7.5 Hz, 2H), 3.13 (s, 2H), 1.78 (s, 3H), 1.10 (t, *J* = 7.5 Hz, 3H); ^13^C-NMR (DMSO-*d_6_*, 125 MHz) *δ*(ppm): 176.0, 167.6, 164.1, 160.7, 156.2, 148.2, 147.1, 145.9, 136.3, 124.5, 122.6, 117.3, 108.3, 107.3, 103.1, 98.3, 93.6, 60.3, 43.1, 24.7, 13.9.

*Benzyl (E)-3-((5-hydroxy-4-oxo-2-phenyl-4H-chromen-7-yl) oxy) but-2-enoate* (**4a**); yellow powder; yield 33.0%; HRMS(ESI): calcd. C_26_H_21_O_6_, *m*/*z* 429.1333 [M + H]^+^, found 429.1324; ^1^H-NMR (DMSO-*d_6_*, 400 MHz) *δ*(ppm): 12.93 (s, 1 H), 8.12 (d, *J* = 6.8 Hz, 2H), 7.74–7.56 (m, 3H), 7.32 (m, 5H), 7.07 (d, *J* = 2.0 Hz, 1H), 6.63 (d, *J* = 2.0 Hz, 1H), 5.10 (s, 1H), 5.07 (s, 2H), 2.46 (s, 3H); ^13^C-NMR (DMSO-*d_6_*, 100 MHz) *δ*(ppm): 182.7, 171.3, 166.1, 164.2, 161.6, 158.7, 157.1, 136.3, 132.5, 130.4, 129.3, 128.5, 128.3, 128.1, 126.8, 108.3, 105.8, 104.4, 100.7, 98.3, 65.3, 17.9. 

*Dibenzyl 3,3′-((4-oxo-2-phenyl-4H-chromene-5,7-diyl) bis (oxy)) (2E,2′E)-bis (but-2-enoate)* (**4b**); white powder; yield 29.8%; HRMS(ESI): calcd. C_37_H_31_O_8_, *m*/*z* 603.2013 [M + H]^+^, found 603.2010; ^1^H-NMR (CDCl_3_, 400 MHz) *δ*(ppm): 7.85 (d, *J* = 6.8 Hz, 2H), 7.52 (d, *J* = 7.6 Hz, 3H), 7.33 (m, *J* = 10H), 7.12 (d, *J* = 2.0 Hz, 1H), 6.71 (d, *J* = 2.0 Hz, 1H), 6.65 (s, 1H), 5.24 (s, 1H), 5.12 (s, 2H), 5.03 (s, 2H), 4.73 (s, 1H), 2.66 (s, 3H), 2.52 (s, 3H); ^13^C-NMR (CDCl_3_, 100 MHz) *δ*(ppm): 175.6, 174.0, 170.6, 167.0, 166.6, 162.3, 158.9, 157.5, 153.0, 136.3, 136.1 133.1, 131.9, 129.2, 128.7, 128.6, 128.5, 128.4, 128.3, 128.2, 126.3, 114.7, 113.3, 108.9, 108.0, 100.2, 95.2, 66.1, 65.8, 18.6, 18.3. 

*Dibenzyl 3,3′-((5-hydroxy-4-oxo-2-phenyl-4H-chromene-6,7-diyl) bis (oxy)) (2E,2′E)-bis (but-2-enoate)* (**4c**); yellow powder; yield 24.2%; HRMS(ESI): calcd. C_37_H_30_O_9_Na, *m*/*z* 641.1782 [M + Na]^+^, found 641.1779; ^1^H-NMR (DMSO-*d_6_*, 400 MHz) *δ*(ppm): 13.17 (s, 1H), 8.16-8.11 (m, 2H), 7.76–7.53 (m, 3H), 7.39-7.23 (m, 10H), 7.18 (s, 1H), 5.14 (s, 1H), 5.04 (s, 1H), 5.01 (s, 2H), 4.96 (s, 1H), 2.43 (s, 3H), 2.42 (s, 3H); ^13^C-NMR (DMSO-*d_6_*, 100 MHz) *δ*(ppm): 182.9, 170.4, 170.3, 166.1, 165.8, 164.5, 153.7, 152.9, 150.5, 136.1, 130.3, 129.2, 128.4, 128.2, 128.1, 128.0, 126.7, 109.4, 105.4, 101.8, 98.0, 94.3, 65.2, 65.1, 17.5, 17.4. 

*Benzyl (E)-3-((5,6-dihydroxy-4-oxo-2-phenyl-4H-chromen-7-yl) oxy) but-2-enoate* (**4d**); yellow powder; yield 16.0%; HRMS(ESI): calcd. C_26_H_20_O_7_Na, *m*/*z* 467.1101 [M + Na]^+^, found 467.1092; ^1^H-NMR (DMSO-*d_6_*, 500 MHz) *δ*(ppm): 13.06 (s, 1H), 11.37 (s, 1H), 8.08 (d, *J* = 6.0 Hz, 2H), 7.62–7.56 (m, 3H), 7.34-7.28 (m, 5H), 7.02 (s, 1H), 6.72 (s, 1H), 5.03 (s, 2H), 4.87 (s, 1H), 2.44 (s, 3H); ^13^C-NMR (DMSO-*d_6_*, 125 MHz) *δ*(ppm): 182.3, 171.2, 166.4, 163.7, 156.6, 154.2, 152.4, 136.3, 132.2, 130.7, 129.2, 128.4, 128.3, 128.0, 126.6, 123.7, 105.0, 104.4, 94.8, 92.9, 65.1, 17.7. 

*Benzyl (E)-3-((5-hydroxy-2-(3-hydroxy-4methoxyphenyl)-4-oxo-4H-chromen-7yl) oxy) but-2-enoate* (**4e**); yellow powder; yield 32.5%; HRMS(ESI): calcd. C_27_H_23_O_8_, *m*/*z* 475.1387 [M + H]^+^, found 475.1375; ^1^H-NMR (DMSO-*d_6_*, 400 MHz) *δ*(ppm): 13.07 (s, 1H), 9.48 (s, 1H), 7.58 (d, *J* = 8.4 Hz, 1H), 7.46 (d, *J* = 1.6 Hz, 1H), 7.32 (s, 5H), 7.08 (d, *J* = 8.4 Hz, 1H), 6.98 (s, 1H), 6.90 (s, 1H), 6.58 (d, *J* = 1.6 Hz, 1H), 5.10 (s, 1H), 5.06 (s, 2H), 3.87 (s, 3H), 2.45 (s, 3H); ^13^C-NMR (DMSO-*d_6_*, 100 MHz) *δ*(ppm): 182.3, 171.3, 166.0, 164.5, 161.6, 158.4, 156.9, 151.5, 146.8, 132.2, 128.4, 128.2, 128.0, 122.6, 119.1, 113.3, 112.1, 108.0, 104.2, 104.1, 100.4, 98.1, 65.2, 55.8, 17.9. 

*Benzyl (E)-3-(4-(3,5,7-trihydroxy-4-oxo-4H-chromen-2-yl) phenoxy) but-2-enoate* (**4f**); yellow powder; yield 23.6%; HRMS(ESI): calcd. C_26_H_20_O_8_Na, *m*/*z* 483.1050 [M + Na]^+^, found 483.1047; ^1^H-NMR (DMSO-*d_6_*, 400 MHz) *δ*(ppm): 12.24 (s, 1H), 10.96 (s, 1H), 10.40 (s, 1H), 7.78–7.76 (m, 2H), 7.32–7.30 (m, 5H), 6.95–6.93 (m, 2H), 6.48 (d, *J* = 2.0 Hz, 1H), 6.23 (d, *J* = 2.0 Hz, 1H), 5.28 (s, 1H), 5.01 (s, 2H), 2.44 (s, 3H); ^13^C-NMR (DMSO-*d_6_*, 100 MHz) *δ*(ppm): 175.0, 170.0, 166.3, 164.5, 161.1, 160.8, 156.8, 156.5, 136.2, 131.0, 130.1, 128.4, 128.3, 128.0, 119.5, 115.9, 104.1, 98.9, 94.8, 94.2, 65.1, 17.6.

*Benzyl (E)-3-(4-(7-(((E)-3-(benzyloxy)-2-methyl-3-oxoprop-1-en-1-yl) oxy)-5-hydroxy-4-oxo-4H-chromen-2-yl) phenoxy)-2-methylacrylate* (**4g**); yellow powder; yield: 64.9%; HRMS(ESI) calcd. C_37_H_31_O_9_, *m*/*z* 619.1963 [M + H]^+^, found 619.1954; ^1^H-NMR (CDCl_3_, 500 MHz) *δ* (ppm): 12.75 (s, 1H), 7.91 (d, *J* = 7.5 Hz, 2H), 7.33 (s, 10H), 7.18 (d, *J* = 8.0 Hz, 2H), 6.68 (d, *J* = 14.5 Hz, 2H), 6.50 (s, 1H), 5.20 (s, 1H), 5.11 (d, *J* = 7.5 Hz, 4H), 5.01 (s, 1H), 2.53 (d, *J* = 11.5 Hz, 6H); ^13^C-NMR (CDCl_3_, 125MHz) *δ* (ppm):182.7, 172.3, 171.2, 167.0, 166.9, 163.7, 162.6, 159.3, 157.3, 156.5, 136.19, 136.1, 128.6, 128.5, 128.4, 128.3, 122.4, 108.6, 106.2, 105.0, 100.1, 99.1, 97.5, 66.0, 65.9, 18.5, 18.3.

*Ethyl (E)-3-((5-hydroxy-4-oxo-2-phenyl-4H-chromen-7-yl) oxy) hex-2-enoate* (**5a**); yellow powder; yield 32.0%; HRMS(ESI): calcd. C_23_H_22_O_6_Na, *m*/*z* 417.1309 [M + Na]^+^, found 417.1305; ^1^H-NMR (CDCl_3_, 400 MHz) *δ*(ppm): 12.79 (s, 1H), 7.89 (dd, *J* = 8.0, 2.0 Hz, 2H), 7.57–7.50 (m, 3H), 6.74 (s, 1H), 6.67 (d, *J* = 2.0 Hz, 1H), 6.49 (d, *J* = 2.0 Hz, 1H), 5.08 (s, 1H), 4.11 (dd, *J* = 2H), 2.94–2.90 (m, 2H), 1.74 (m, *J* = 15.2, 7.6 Hz, 2H), 1.23 (t, *J* = 7.2 Hz, 3H), 1.05 (t, *J* = 7.2 Hz, 3H); ^13^C-NMR (CDCl_3_, 100 MHz) *δ*(ppm): 182.9, 174.4, 166.9, 164.7, 162.6, 159.7, 157.5, 132.3, 131.0, 129.3, 126.5, 108.6, 106.3, 105.0, 100.1, 99.2, 60.0, 32.9, 20.9, 14.4, 13.9. 

*Ethyl (E)-3-((7-hydroxy-4-oxo-2-phenyl-4H-chromen-5-yl) oxy) hex-2-enoate* (**5b**); yellow powder; yield 40.0%; HRMS(ESI): calcd. C_23_H_23_O_6_, *m*/*z* 395.1489 [M + H]^+^, found 395.1482; ^1^H-NMR (DMSO-*d_6_*, 400 MHz) *δ*(ppm): 8.26–8.01 (m, 2H), 7.56 (d, *J* = 7.2 Hz, 3H), 6.98 (d, *J* = 2.2 Hz, 1H), 6.74 (s, 1H), 6.49 (d, *J* = 2.2 Hz, 1H), 4.49 (s, 1H), 3.94 (dd, *J* = 14.4, 7.2 Hz, 2H), 2.94–2.90 (m, 2H), 1.76 (m, *J* = 2H), 1.09–0.99 (m, 6H), 1.00 (t, *J* = 7.2 Hz, 3H); ^13^C-NMR (DMSO-*d_6_*, 100 MHz) *δ*(ppm): 176.1, 174.6, 166.2, 162.6, 160.7, 158.8, 151.8, 131.6, 130.9, 129.1, 126.2, 109.5, 108.6, 107.7, 101.5, 93.7, 32.6, 59.1, 19.9, 14.1, 13.8. 

*Ethyl (E)-3-((7-(((Z)-1-ethoxy-1-oxohex-2-en-3-yl) oxy)-6-hydroxy-4-oxo-2-phenyl-4H-chromen-5-yl) oxy) hex-2-enoate* (**5c**); yellow powder; yield 7.0%; HRMS(ESI): calcd. C_31_H_35_O_9_, *m*/*z* 551.2276 [M + H]^+^, found 551.2260; ^1^H-NMR (CDCl_3_, 400 MHz) *δ*(ppm): 8.40 (s, 1H), 7.83 (d, *J* = 6.4 Hz, 2H), 7.55–7.48 (m, 3H), 7.11 (s, 1H), 6.60 (s, 1H), 4.90 (s, 1H), 4.70 (s, 1H), 4.10–4.03 (m, 4H), 3.05–3.02 (m, 2H), 2.93–2.90 (m, 2H), 1.87–1.72 (m, 4H), 1.21–1.16 (m, *J* = 6H), 1.07–1.02 (m, 6H); ^13^C-NMR (CDCl_3_, 125 MHz) *δ*(ppm): 176.5, 175.5, 173.6, 167.3, 167.1, 162.5, 156.0, 154.6, 144.1, 132.0, 131.6, 131.0, 129.2, 126.3, 111.4, 107.8, 102.7, 95.6, 94.5, 60.1, 59.9, 33.6, 33.1, 21.2, 20.7, 14.4, 14.2. 

*Ethyl (E)-3-((5,6-dihydroxy-4-oxo-2-phenyl-4H-chromen-7-yl) oxy) hex-2-enoate* (**5d**); yellow powder; yield 11.2%; HRMS(ESI): calcd. C_23_H_21_O_7_, *m*/*z* 409.1293 [M − H]^−^, found 409.1289; ^1^H-NMR (DMSO-*d_6_*, 500 MHz) *δ*(ppm): 13.03 (s, 1H), 11.36 (s, 1H), 8.07 (d, *J* = 7.5 Hz, 2H), 7.61–7.56 (m, 3H), 7.01 (s, 1H), 6.70 (s, 1H), 4.74 (s, 1H), 3.98 (dd, *J* = 14.5, 6.0 Hz, 2H), 2.86 (t, *J* = 7.0 Hz, 2H), 1.71 (dd, *J* = 15.0, 7.5 Hz, 2H), 1.10 (t, *J* = 7.0 Hz, 3H), 1.00 (t, *J* = 7.5 Hz, 3H); ^13^C-NMR (DMSO-*d_6_*, 100 MHz) *δ*(ppm): 182.3, 173.8, 166.3, 163.6, 156.7, 154.1, 152.4, 132.1, 130.7, 129.2, 126.5, 123.7, 104.9, 104.4, 94.7, 92.8, 59.1, 32.3, 20.4, 14.1, 13.5. 

*Diethyl 3,3′-((5-hydroxy-4-oxo-2-phenyl-4H-chromene-6,7-diyl) bis (oxy)) (2E,2′E)-bis (hex-2-enoate)* (**5e**); yellow powder; yield 14.8%; HRMS(ESI): calcd. C_31_H_34_O_9_Na, *m*/*z* 573.2095 [M + Na]^+^, found 573.2106; ^1^H-NMR (CDCl_3_, 500 MHz) *δ*(ppm): 12.96 (s, 1H), 7.89 (d, *J* = 7.5 Hz, 2H), 7.60–7.53 (m, 3H), 6.76 (d, *J* = 9.0 Hz, 1H), 5.03 (s, 1H), 4.85 (s, 1H), 4.13–4.06 (m, 4H), 2.93–2.87 (m, 4H), 1.77–1.69 (m, 4H), 1.21 (dd, *J* = 15.2, 7.6 Hz, 6H), 1.06–1.00 (m, 6H); ^13^C-NMR(CDCl_3_, 125 MHz) *δ*(ppm): 182.9, 174.0, 173.8, 167.1, 166.7, 165.2, 154.1, 153.8, 151.8, 132.5, 130.9, 129.4, 128.4, 126.6, 109.5, 105.9, 100.7, 98.7, 94.5, 60.0, 59.7, 33.1, 32.8, 21.0, 20.9, 14.4, 14.0, 13.9. 

*(E)-3-((5-hydroxy-4-oxo-2-phenyl-4H-chromen-7-yl) oxy) but-2-enoic acid* (**6a**); yellow powder; yield 69.4%; HRMS(ESI): calcd. C_19_H_13_O_6_, *m*/*z* 337.0718 [M + H]^+^, found 337.0711; ^1^H-NMR (DMSO-*d*_6_, 500 MHz) *δ*(ppm): 12.93 (s, 1H), 8.14 (d, *J* = 6.4 Hz, 2H), 7.76–7.58 (m, 3H), 7.15 (s, 1H), 7.05 (d, *J* = 2.0 Hz, 1H), 6.61 (d, *J* = 2.0 Hz, 1H), 5.09 (s, 1H), 2.41 (s, 3H); ^13^C-NMR (DMSO-*d*_6_, 125 MHz) *δ*(ppm): 182.8, 169.7, 167.6, 164.3, 161.7, 159.3, 157.2, 132.6, 130.6, 129.4, 126.9, 108.1, 105.9, 104.2, 100.7, 100.3, 17.7. 

*2-(2-methyl-5-(3,5,7-trihydroxy-4-oxo-4H-chromen-2-yl) benzo[d] [1,3] dioxol-2-yl) acetic acid* (**6b**); yellow powder; yield 43.7%; HRMS(ESI): calcd. C_19_H_15_O_9_, *m*/*z* 387.0711 [M + H]^+^, found 387.0707; ^1^H-NMR (DMSO-*d_6_*, 500 MHz) *δ*(ppm): 12.40 (s, 1 H), 7.73 (dd, *J* = 10.5, 1.5 Hz, 1H), 7.61 (d, *J* = 1.5 Hz, 1H), 7.03 (d, *J* = 10.5 Hz, 1H), 6.46 (d, *J* = 2.5 Hz, 1H), 6.19 (d, *J* = 2.5 Hz, 1H), 3.02 (s, 2H), 1.78 (s, 3H); ^13^C-NMR (DMSO-*d*_6_, 125 MHz) *δ*(ppm): 217.5, 176.0, 172.1, 169.2, 164.1, 160.7, 156.2, 148.2, 147.1, 146.0, 136.2, 124.8, 122.9, 117.6, 108.4, 107.3, 103.1, 98.3, 93.7, 24.5, 21.1. 

### 2.3. Anti-Cancer Activity Assay

#### 2.3.1. Cell Lines and Cell Culture

The human leukemia cell lines K562 and HEL, the non-small cell lung cancer A549, and the prostate cancer cell line PC3 were obtained from the Key Laboratory of Chemistry for Natural Product of Guizhou Province and Chinese Academy of Science (Guiyang, China). Cells were cultured as a monolayer in DMEM (Hyclone, Germany), supplemented with 10% heat-inactivated research-grade fetal bovine serum (Hyclone, Germany) and penicillin/streptomycin (Sigma, St. Louis, MO, USA) at 37 °C in a humidified atmosphere, containing 5% CO_2_.

#### 2.3.2. Cytotoxic Activity Assay

The cytotoxic activity of compounds was measured by MTT assay, using adriamycin as the positive control. Cells were seeded in 96-well microculture plates at a density of 1 × 10^4^ cells/well and left to adhere for 24 h. The cells were then exposed to different concentrations of the compounds for 48 h. MTT was added to each well at a final concentration of 0.5 mg/mL and cells were incubated at 37 °C for an additional 4 h. The medium was then discarded and 200 μL Tris-DMSO solution was added to each well. The dark blue formazan crystals formed were dissolved by slight shaking, and the absorbance was measured at 490 nm, using an ELISA plate reader.

#### 2.3.3. Statistical Analysis

The data were analyzed using SPSS 18.0 and reported as mean ± standard deviation (SD) of the number of experiments indicated. For all the measurements, one-way ANOVA followed by a Student’s *t*-test was used to assess the statistical significance of the differences between each group. The statistical significance of the difference between two groups was assessed using the LSD method. *p* < 0.05 was considered to indicate statistical significance. The data are presented as the mean ± standard error of the mean (SEM) of three assays. 

## 3. Results and Discussion

### 3.1. Synthesis

The optimal reaction conditions were established (Table 1, entry 6). The substrate scope was further extended to other types of allenes (Figure 1). As expected, various substituent groups led to a variety of products. When using either the monoallene (ethyl ester/benzyl ester) or diallene, the Pd catalyst efficiently promoted this C-O functionalization. Intriguingly, the application of triethylamine as a base instead of potassium carbonate was demonstrated to be more efficient for target product purification (**4e**, **4f**). However, completion of the reaction is hard to achieve. Starting flavonoids with OH groups at positions 3′ and 4′ easily generated the 1,3-dioxolane byproducts, instead of the desired ones. The key intermediates **1a**–**1c** were prepared according to reported methods [18,19,20].

Under optimal conditions, as illustrated in Figure 2, a variety of flavonoid substrates was investigated. For instance, flavones and flavonols were well tolerated, providing ample opportunities for further anti-cancer activity evaluation of the derivatives. We found that the property of flavonoid and the substituted phenolic hydroxyl groups had a great impact on the formation of the desired products. Notably, it was more challenging to separate the analog products when more than two hydroxyl groups were presented. Finally, the target products (**3a**–**3i**, **4a**–**4g**, **5a**–**5e**, **6a**, **6b**) were successfully purified by silica gel column chromatography. The configuration of the C=C bond in **3a** and **4a** was determined by the NOE (Nuclear Overhauser Effect) study. The structures of final compounds were characterized by ^1^H-, ^13^C-NMR, and MS spectrometry.

### 3.2. Anti-Cancer Activity

The anti-tumor activity of all derivatives in vitro was summarized in Table 2. Results showed that the inhibitory effect of compound **4e** on the growth of K562 was six times stronger (*p* < 0.01) than that of the substrate, diosmetin, which indicated that the anti-tumor activity of the diosmetin derivative was associated with the more lipophilic benzyl ester than -OCH_3_ moiety at R_6_ position of the substrate. Compounds **5e**, **3h**, and **6b** showed better activity (*p* < 0.01) on the proliferation of HEL than that of the substrates, baicalein, kaempferol, and quercetin, respectively. These results suggested that the anti-leukemia activity may be improved by increasing steric hindrance, such as by introducing propyl and 1,3-dioxolane groups onto baicalein. Compound **6b** exhibited an enhanced inhibitory activity on the growth of leukemia cells (K562 and HEL) and non-small cell lung cancer A549 compared with those of **3i**, which can be speculated as the enhanced water solubility associated with the carboxyl group. We then synthesized compounds **3e**, **4b**, and **4c** to improve the anti-cancer activity (*p* < 0.01) on the proliferation of PC3 compared with the substrates, diosmetin, chrysin, and baicalein, respectively. The results suggested that introducing an ethyl and benzyl esters could increase their anti-cancer activity.

## 4. Conclusions

Twenty-three flavonoidal ether derivatives were synthesized without the steps of introducing protective groups followed by deprotection. The in vitro tumor growth inhibitory activities of all the derivatives were assayed using four human cancer cell lines, K562, A549, HEL, and PC3. In general, compounds containing 1,3-dioxolane, such as **6b**, possessed broad-spectrum inhibitory activity against the above four cancer cells. Preliminary structure–activity relationship studies indicated that the position-7 substituents were essential for the cytotoxicities of the derivatives. These results provided new insight into developing flavonoid-derived anti-cancer agents.

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
