# Peer review of "Design and Synthesis of Flavonoidal Ethers and Their Anti-Cancer Activity In Vitro"

_molecules, 2019, doi:10.3390/molecules24091749_

Round 1
Reviewer 1 Report
The content of the Ms content information to develop a new lead for producing a new drog that kill cancer cell. I recomend for publication. Detail comments are attached.

Author Response
Thanks for your consideration, and we already med additional inprovement for our revised manuscript.
Reviewer 2 Report
In the paper "Design and synthesis...." Heng Luo et al. sutdied the substrate scope of the reaction to various types of allene partners providing the possibility to change the reactivity pattern. Since most flavonoids have not been developed as clinical drugs due to poor bioavailability, they report a strategy for increasing the drug-ability of flavonoids by constructing C(sp2)-O bonds and stereo- as well as regio-selective alkenylation of hydroxyl groups of flavonoids with ethyl-2,3-butadienoate allenes. 23 modified flavonoid derivatives were designed, synthesized and evaluated for their anti-cancer activities.
The paper is well written, the experimental details are clear.
However, there is one point, which should be taken into consideration, before tghe paper can be accepted for publication in Molecules.
Namely, the new compounds are characterized only by 1H, 13C and ESI-MS. The athors should pride for all new compounds either the microanalyses for C, and H, or HRMS!
Author Response
Thanks for your comments, and we provided the HRMS data for all the new derivatives.
Reviewer 3 Report
This paper is limited in many sections:
1) The English language is very poor. Many sentences are not meaningful and many word are misused as example “these efforts were agnostic with respect to”
2) The most of references are inadequate, old and/or not relevant to this paper. As non-exhaustive examples reference 1 does not deal with the discovery of flavonoids but is a brief celebrative note of Albert Szent-Gyorgyi, references 7 and 8 do not deal with improvement of bioavailability or toxicity profile restriction but with biological activity of flavonoids, references 18-23 deal with chemical probes in proteome research and not with minimization of off-target reactivity-linked toxicities and so on. Some references are repeated twice see 12b and 13. The references on allenes reactivity are too much, less relevant to this manuscript and not recent as indicated in the text.
3) The section 2.2.2 is bad written, as non-exhaustive examples the ligand is always the same, table 1 entry 1 and 3 are mismatched, 11.2 % yields should not be considered as trace
4) The experimental section must be separated from the discussion on chemistry. Furthermore, how was added allene into a sealed tube? and 1 mmol is not large scale as compared to 0.5 mmol
5) Schemes and tables data do not much, compound 3b, 3c, 4c, 4d, 5c, 5d and 5e are not derived from 2a as indicated in the scheme
6) The discussion of biological data is highly questionable and some of the reported statements are not supported by experimental data. As non-exhaustive examples no water solubility has been measured and no correlation occurs on 4a and 6a biological data
In conclusion this is not the quality and relevance required for publication on Molecules
Author Response
1) The English language is very poor. Many sentences are not meaningful and many word are misused as example “these efforts were agnostic with respect to”
Response: Thanks for your comments, we have corrected all the language issues in the whole manuscript and checked several times by a Canadian expert.
2) The most of references are inadequate, old and/or not relevant to this paper. As non-exhaustive examples reference 1 does not deal with the discovery of flavonoids but is a brief celebrative note of Albert Szent-Gyorgyi, references 7 and 8 do not deal with improvement of bioavailability or toxicity profile restriction but with biological activity of flavonoids, references 18-23 deal with chemical probes in proteome research and not with minimization of off-target reactivity-linked toxicities and so on. Some references are repeated twice see 12b and 13. The references on allenes reactivity are too much, less relevant to this manuscript and not recent as indicated in the text.
Response: Thanks for your comments, we have renewed most of the old references and removed reference 1 as well as some inappropriate ones according to your suggestion.
3) The section 2.2.2 is bad written, as non-exhaustive examples the ligand is always the same, table 1 entry 1 and 3 are mismatched, 11.2 % yields should not be considered as trace
Response: Thanks for your comments. We have revised this section, and changed the “Interestingly, trace amounts of the expected product were detected at lower temperatures (Table 1, entry 17)” into “Interestingly, trace amounts of the expected product were detected at lower temperatures (Table 1, entry 17)” in the revised manuscript.
4) The experimental section must be separated from the discussion on chemistry. Furthermore, how was added allene into a sealed tube? and 1 mmol is not large scale as compared to 0.5 mmol
Response: Thanks for your comments. First of all, we have separated the experimental section from the discussion on chemistry. Secondly, the allenes was added into a special “sealed tube” which can be sealed using a Teflon® thread cap without flame. For the last question, we have removed the words “large scale” in the revised manuscript.
5) Schemes and tables data do not much, compound 3b, 3c, 4c, 4d, 5c, 5d and 5e are not derived from 2a as indicated in the scheme
Response: Maybe there is a misunderstanding here. We have chosen eight different flavones as starting materials to be modified with allenes but only took 2a as an example to describe the procedure.
6) The discussion of biological data is highly questionable and some of the reported statements are not supported by experimental data. As non-exhaustive examples no water solubility has been measured and no correlation occurs on 4a and 6a biological data
Response: We have checked the discussion of biological data and statements and deleted the relation between the biological activity and water solubility of 4a and 6a in the revised manuscript.
Round 2
Reviewer 3 Report
The manuscript quality has been improved. However some revisions are needed.
The instrument used to obtain HRMS must be indicated.
The method used to seal reaction tubes and add further reactants must be explained in the manuscript.
Schemes and tables are still not clear
Author Response
1. The manuscript quality has been improved. However some revisions are needed.
Response: Thanks for your comments, we made further revisions regarding to the editing issues.
2. The instrument used to obtain HRMS must be indicated.
Response: Thanks for your comments, we have revised/added the information of the instrument used to obtain HRMS in the revised manuscript.
3. The method used to seal reaction tubes and add further reactants must be explained in the manuscript.
Response: Thanks for your comments, we have reorganized the related paragraph for better indicating the "sealed reaction tubes".
4. Schemes and tables are still not clear.
Response: Thanks for your comments, we have reorganized fig. 2 in the revised manuscript.